# Real-World Data Analysis of Pregnancy-Associated Breast Cancer at a Tertiary-Level Hospital in Romania

**DOI:** 10.3390/medicina56100522

**Published:** 2020-10-06

**Authors:** Anca A. Simionescu, Alexandra Horobeț, Lucian Belaşcu, Dragoş Mircea Median

**Affiliations:** 1Department of Obstetrics and Gynecology, Carol Davila University of Medicine and Pharmacy, Filantropia Clinical Hospital, 011171 Bucharest, Romania; 2Department of International Business and Economics, The Bucharest University of Economic Studies, 010374 Bucharest, Romania; alexandra.horobet@rei.ase.ro; 3Department of Management, Marketing and Business Administration, Lucian Blaga University of Sibiu, 550024 Sibiu, Romania; lucian.belascu@ulbsibiu.ro; 4Gynecologic Oncology Department, Filantropia Clinical Hospital Bucharest, 011171 Bucharest, Romania

**Keywords:** breast cancer, pregnancy-associated breast cancer, Romania

## Abstract

*Background and objectives:* Breast cancer is among the most common cancer types encountered during pregnancy. Here, we aimed to describe the characteristics, management, and outcomes of women with pregnancy-associated breast cancer at a tertiary-level hospital in Romania. *Material and Methods:* We retrospectively and prospectively collected demographic, oncological, and obstetrical data for women diagnosed with cancer during pregnancy, and who elected to continue their pregnancy, between June 2012 and June 2020. Complete data were obtained regarding family and personal medical history and risks factors, cancer diagnosis and staging, clinical and pathological features (including histology and immunohistochemistry), multimodal cancer treatment, pregnancy management (fetal ultrasounds, childbirth, and postpartum data), and infant development and clinical evolution up to 2020. Cancer therapy was administered following national guidelines and institutional protocols and regimens developed for non-pregnant patients, including surgery and chemotherapy, while avoiding radiotherapy during pregnancy. *Results:* At diagnosis, 16.67% of patients were in an advanced/metastatic stage, while 75% were in early operable stages. However, the latter patients underwent neoadjuvant chemotherapy rather than up-front surgery due to aggressive tumor biology (triple negative, multifocal, or HER2+). No patient achieved complete pathological remission, but only one patient relapsed. No recurrence was recorded within 12 months among early-stage patients. *Conclusions:* In this contemporary assessment of real-world treatment patterns and outcomes among patients with pregnancy-associated breast cancer, our findings were generally consistent with globally observed treatment outcomes, underscoring the need for a multidisciplinary team and reference centers.

## 1. Introduction

Breast cancer (BC) is the most frequent female cancer worldwide, and the leading cause of cancer-related death among women [1]. It is also one of the most common types of cancer encountered during pregnancy [2,3]. Data regarding the global burden of disease between 1990–2017 among Romanian women of fertile age (15–49 years) reveal that although the number of BC-related deaths has decreased, the importance of BC as a cause of mortality has increased by 38.51%. The estimated probability of pregnancy-associated breast cancer (PABC) is 1.29/100,000 among women aged 15–49 years, mainly due to pregnancy at an older age. We expect 59 new cases of PABC per year in Romania (Appendix A).

PABC guidelines were first introduced in 1999 [4]. Since then, substantial progress has been made regarding management and follow-up care. In 2017, Romania joined the International Network on Cancer, Infertility, and Pregnancy (INCIP). Globally, PABC accounts for 0.2–7% of BC cases [5,6,7]. PABC is not a single disease, but rather several diseases distinguished by the biological effects of pregnancy, variations in serum hormone levels [8], extracellular matrix modifications, and distinctive gene expression patterns in mammary epithelial cells [9]. Research over recent decades has focused on identifying potential PABC biomarkers [10]. Cancer screening can be performed in pregnant women by using a non-invasive prenatal screening test (NIPT) involving plasma cell-free DNA sequencing [11]. Higher maternal serum levels of cell-free DNA (cfDNA) are detected in cases of PABC with a euploid fetus, likely due to both harbored fragments of cell-free fetal DNA and cell-free tumoral DNA [12,13,14]. It has been hypothesized that hypertension during pregnancy or preeclampsia may decrease the risk of breast cancer development [15,16,17].

Traditionally, BC is considered PABC when the disease is diagnosed during pregnancy or in the first postpartum year [18,19]. Breast cancer diagnosed between 1 and 10 years after childbirth in women under 45 years of age seems to be a different entity, carrying increased risks of distant metastases and death, which can be explained at the molecular level by the phenomenon of breast involution [20,21].

In the recent past, the diagnosis of breast cancer during pregnancy has often been followed by induced abortion, as many doctors, patients, and patients’ relatives have been concerned about the poor prognosis and high risk of maternal death or fetal adverse reaction following the disease or treatments. However, therapeutic abortion is no longer routinely recommended upon PABC diagnosis. With the application of a standard treatment, experts consider the prognosis of cancer during pregnancy to be similar to that in non-pregnant patients [22]. Studies have shown that BC diagnosis during pregnancy may be delayed due to the fact that this pathology is not specific to pregnancy, and because of the physiological increase of mammary gland size [23]. Staging and treatment regimens for PABC are similar in non-pregnant patients. Surgery and sentinel lymph nodes biopsy (SNB) are allowed during pregnancy, despite controversial findings regarding the safety and accuracy of the results [24,25]. SNB in pregnant breast cancer patients appears to be safe and accurate using either methylene blue dye or technetium 99 m (99 mTc) sulfur colloid and gamma probe according to a retrospective study [25]. However, recommendations from the American Society of Clinical Oncology (ASCO) still state that pregnant patients should not undergo sentinel lymph node biopsy (SLNB), based on cohort studies and informal consensus [26]. In contrast with ASCO, The National Comprehensive Cancer Network NCCN concluded that insufficient data exist on which to base recommendations regarding the use of SLNB in pregnant women and SLNB should not be offered to pregnant women under 30 weeks of gestation [27].

During surgery, pregnant women are considered at increased risk of hypoxemia, desaturation, and thrombosis. The more advanced the term, the more difficult it is to intubate, especially obese pregnant patients. The patient should be positioned as far as possible in the left lateral position to avoid aortocaval compression and hypotension that decreases uterine vascularization. Intraoperative cardiotocography for fetal heart rate monitoring may indicate a reduced fetal heart rate variability during general anesthesia. Administration of intravenous tocolytic drugs, e.g., betamimetics or atosiban may be used for the inhibition of uterine contractility [28,29].

In the second or third trimester, adjuvant or neoadjuvant chemotherapy (CMT) regimens include anthracycline-based treatment, followed by taxanes (paclitaxel and docetaxel) [30,31,32,33], and the main known fetal risks include intrauterine growth restriction and a small-for-gestational age newborn. During the postpartum period, the recommended treatment includes anti-HER-2 targeted therapy for HER-2+ tumors, and endocrine therapy for oestrogen-receptor (ER)-positive tumors. Radiotherapy is considered relatively safe during the first and second trimesters of pregnancy, although this is based on theoretical assumptions and few experiences [34,35]. In utero radiation exposure of >10 mGy has been associated with microcephaly, childhood cancers, and mental retardation [36,37,38,39,40]. Radiotherapy during pregnancy remains controversial. Some guidelines suggest that it can be safely used in the first or second trimester of pregnancy [19], while Romanian guidelines contraindicate radiotherapy for BC during pregnancy [41].

Upon diagnosis of cancer during pregnancy, selection of the optimal treatment approach requires a multidisciplinary team of healthcare professionals—including an obstetrician, a specialist in maternal–fetal medicine, a medical oncologist, an oncology surgeon, a pediatrician, a geneticist, a psycho-oncologist, and a radiotherapist. A multidisciplinary team can advise the patient and her family about the available treatment regimens, the potential risks for both the mother and the fetus, the prognosis, and the necessity of a long-term follow-up.

In the present study, we aimed to describe the characteristics, management, and follow-up care of women with PABC at a tertiary-level hospital in Romania.

## 2. Materials and Methods

We performed retrospective and prospective collection of data regarding women who were diagnosed with cancer during pregnancy and who elected to continue their pregnancy, between June 2012 and June 2020. All subjects gave their informed consent for inclusion before they participated in the study. The study was conducted in accordance with the Declaration of Helsinki, and the protocol was approved by the Hospital Ethics Committee (3299/20 March 2020). The “Filantropia” Clinical Hospital in Bucharest has departments specialized in gynecological oncology—including medical oncology, pathology, radiology, obstetrics, and maternal and fetal medicine—as well as a tertiary-level maternity ward equipped to provide neonatal resuscitation, and a counselling psychologist. Since 2015, a weekly multidisciplinary team (including gynecologic surgeons, radiotherapists, oncologists, and geneticists) meets to provide protocols for diagnoses and treatment plans (Figure 1) for any cancer patient.

Data were retrieved from the hospital registries and patient files. We obtained complete data regarding family and personal medical history and risks factors; cancer diagnosis and staging; clinical and pathological features, including histology and immunohistochemistry (IHC); and multimodal cancer treatment. We also retrieved data about pregnancy management, including fetal ultrasound evaluations, childbirth, and the postpartum period. Finally, we collected data regarding infant development and clinical evolution after birth, up to 2020. In Romania, the National Insurance House does not reimburse genetic testing for a tumor genome, for cancer risk or NIPT in pregnancy. Thus, cancer genetic tests were available for few patients. Diagnoses were mostly suspected based on patient self-examination, rather than a scheduled medical exam. Staging was generally established by ultrasound assessment—with mammography or other imaging studies recommended only for postpartum patients. Breast cancer was staged according to the American Joint Committee on Cancer (AJCC) Tumor-Node-Metastasis (TNM)-staging system [42,43]. Immunohistochemistry assessment was used to categorize breast cancer into the following subtypes: Luminal A, Luminal B HER2−, Luminal B HER2+, HER2+ (non-luminal), and triple-negative [44]. Data were analyzed descriptively.

Surgery was performed regardless of the gestational age of the pregnancy. For all cases, axillary staging was performed by axillary lymph node dissection (ALND). For tumors greater than 2 cm (determined by ultrasound), we offered neoadjuvant chemotherapy (NACT) starting in the second trimester of pregnancy. Systemic therapy was performed using the same regimens and schedules as in non-pregnant patients, including anthracyclines (epirubicin or doxorubicin/cyclophosphamide) and taxane-based chemotherapy (paclitaxel, docetaxel). No dose-dense regimen was used. Chemotherapy was stopped no later than 35 weeks of gestation, according to available international guidelines [19], to avoid the delivery occurring during the time-frame of maternal pancytopenia, which could lead to fetal complications (hematologic toxicities or infection) [19]. Due to the different gestational ages at the start of systemic treatment, taxanes were usually given postpartum. Patients with HER2+ disease received anti-HER2 targeted therapy (i.v. or s.c. trastuzumab, pertuzumab, or trastuzumab emtansine) after delivery, according to the drugs’ label. The delayed administration of anti-HER2 therapy, for a few weeks, after delivery, is unlikely to impact the outcome negatively. The four-year follow-up of the HERA study suggested that patients who received delayed treatment with Herceptin at a median of 2 years following chemotherapy had a lower risk of relapse compared to patients who remained in the observational, non-treated group [45]. For patients with luminal disease, postpartum endocrine therapy disease was recommended. When indicated, radiotherapy was administered after delivery.

These high-risk pregnancy management options included fetal ultrasound assessment for structural abnormalities and fetal growth, including Doppler evaluation (umbilical artery Doppler assessment, *ductus venosus (DV)*, middle cerebral artery, cerebroplacental ratio, uterine arteries) and amniotic fluid level measurements after systemic chemotherapy, according to our local protocols. Maternal and fetal echocardiography monitoring of the serial left ventricular ejection fraction was performed before and after anthracycline-based chemotherapy administration. Corticosteroid treatment for lung maturation was offered at between 28 and 32 gestational weeks. Birth induction was rare and was mostly performed for obstetric indications (breech presentation, intrauterine growth restriction with abnormal DV blood flow, placenta previa, first twin in non-cephalic presentation), or if the maternal condition had deteriorated. Vaginal birth was preferred and recommended after 37 weeks. Breastfeeding was contraindicated while a mother was undergoing systemic treatment or radiation therapy. Lactation was suppressed when continued cancer treatment was needed. Newborns were clinically evaluated, subjected to echocardiography, and, if the mother agreed, monitored every six months for physical and neurologic development and age-appropriate scores on echocardiography evaluation.

## 3. Results

During the studied time period, 13 cases of PABC were managed at our hospital, with an average of 1.62 new cases per year. Of these 13 cases, 12 are included in our analysis. One was excluded because, while the patient underwent surgery, oncological treatment was postponed due to severe acute respiratory syndrome coronavirus 2 (SARS-CoV-2) infection. The 12 analyzed patients with PABC gave birth to 13 babies (one pair of twins). During the investigation period, 30,355 children were born in the Department of Obstetrics, and 295 new cases of breast cancer in women of childbearing age (16–49 years) were treated in our Medical Oncology Department. Among age-matched women, the incidence rate of PABC was 4.4%.

Table 1 presents the patients’ characteristics. The mean age at diagnosis was 35 years (range, 24–38 years), with 91.67% of patients being over 30 years old. Two-thirds of the cases were among patients 36–40 years of age. BC was diagnosed during the second trimester in 41.67% of cases, and during the third trimester in 33.33% of cases. Two patients were multigravida, five (41.67%) primigravida, and five secundigravida. In all cases of multigravidity, <5 years had passed since the previous pregnancy. The lump was more frequently found in the right breast. Four patients had a positive family history of cancer. None of the patients smoked or drank alcohol.

Table 2 presents the cancer characteristics, management, and issues. Disease-free survival (DFS) was defined as the interval (in months) from the date of primary surgery to the first locoregional recurrence or distant metastasis, as of 1 August 2020. Overall survival (OS) was the interval (in months) from the date of diagnosis to breast cancer-related death. The majority of cases (75%) were diagnosed in early stages. However, due to aggressive tumor biology, treatment was started with NACT in 8 of 12 patients. Among the analyzed cases, two were triple negative breast cancer (TNBC), five were HER2+ disease, one was multifocal multicentric disease, and two patients presented in the metastatic stage. Among the 12 patients, 5 underwent treatment during pregnancy, which included anthracycline-based chemotherapy, followed by treatment with taxanes. Taxanes were most commonly administered in the postpartum period. Conventional adverse events were recorded, including nausea (grade 1); hematologic toxicities, such as anemia and leukopenia (grade 1); alopecia (grade 1); and sensitive neuropathy (grade 2). No experienced adverse events required that treatment be stopped or postponed.

Patient No. 9 had TNBC and exhibited non-pathologic complete response (non-pCR), and received capecitabine in the adjuvant setting. No patient achieved pCR with NACT. The same regimens were used in the adjuvant setting for the patients with upfront surgery if chemotherapy was considered.

For the patients with HER2+ disease, trastuzumab was offered in one case, and trastuzumab plus pertuzumab in two cases. One patient with non-pCR was offered trastuzumab emtansine in the adjuvant setting. For the other patients, treatment was followed with trastuzumab alone. Only one patient (No. 3) did not receive anti-HER2 treatment. After four cycles of AC, she opted for surgery. On postoperative day 5, the patient∙s general status worsened. She was diagnosed with metabolic encephalopathy and brain edema and died six days after the surgical intervention. The family refused the necropsy, so we do not have a conclusion for this case.

One patient (No. 5) received taxanes (docetaxel) and trastuzumab in the metastatic setting (lung, liver, and bone). She achieved a partial response, but she left the country. After disease progression, her treatment was switched to capecitabine, trastuzumab, and pertuzumab. However, after a few months, we were unable to contact her anymore. She was the only patient lost to follow-up.

The most eventful follow-up was for patient No. 2, who was diagnosed at 32 years old during her second pregnancy. She received NACT and underwent radical mastectomy. After four years of adjuvant endocrine therapy (tamoxifen and goserelin), the patient developed bone metastasis. Treatment was changed to fulvestrant and the bone-modifying agent zoledronic acid. Two years later, liver metastasis occurred, and the patient resumed chemotherapy with capecitabine, gemcitabine, and PLD. Meanwhile, several brain metastases were diagnosed and radiotherapy performed. In May 2020, the patient was diagnosed with contralateral BC, and a simple mastectomy was performed. The pathology report showed a triple-negative disease in contrast with the initial luminal B tumor.

Except for two patients (No.5 and No.10) who were in the metastatic stage, all patients underwent surgery, including radical modified mastectomy and ALND. A prophylactic contralateral mastectomy was also performed in four cases: in one patient with BRCA1 mutation, one with Rad50 mutation (variant of unknown significance (VUS), and two with no or unknown gene mutation. Five patients (including the four with prophylactic contralateral mastectomy) underwent breast reconstruction—four immediately after surgery, one delayed.

Patient No. 6 died immediately postpartum. This patient was a 37-year-old multiparous woman with invasive ductal carcinoma G3, ER+, PgR+, HER2−, T4bN3M1 with skin metastases. She had presented with a lump since her previous pregnancy and had been hospitalized due to general condition degradation [46].

After birth, two patients (No. 11 and No. 12) received anti-HER2-targeted therapy with trastuzumab and pertuzumab, in a neoadjuvant setting, but no pCR was achieved. They went on with an anti-HER2 treatment in an adjuvant setting, but the short follow-up did not allow any DFS consideration.

Adjuvant radiotherapy was offered to three patients. Hormonal therapy was offered to all six patients with ER+ or PgR+ disease.

No patients were diagnosed with obstetrical complications of pregnancy, structural malformation, or restrictive intrauterine growth. No cases involved premature rupture of membranes after NACT. All patients gave birth by cesarean section, 50% due to obstetrical indications, and 50% due to maternal request. All infants were in the 10th to 90th percentile for weight, with Apgar scores of ≥8 at 1, 5, and 10 min. For the two patients with a deteriorated general condition, induction of labor or cesarean section was indicated at 33 weeks, and these patients died in the postpartum period. Their babies were admitted in NICU due to prematurity.

None of the patients in this series wanted an additional pregnancy after completing the treatment. Two-thirds of the patients are currently free of disease with a follow-up of 5–101 months. All infants have exhibited healthy development.

## 4. Discussion

In this study, we performed a contemporary analysis of real-world treatment patterns and outcomes among cases of PABC in a tertiary-level hospital in Romania. This hospital treated an average of two PABC cases per year. Although this may seem small, it suggests that approximately 1568 women of fertile age may suffer PABC throughout the European Union each year (Appendix A). We found that PABC incidence was increased in association with increasing maternal age, with a mean age at diagnosis of 35 years, and 91.67% of patients over 30. We also found that PABC was associated with multiparity and shorter time (≤5 years) between pregnancies. BC is commonly associated with genetic, environmental, reproductive, and lifestyle factors. First childbirth at a young age (under 25 years) [47] reduces the risk of luminal subtype tumors [48,49] and might provide a lesser degree of protection against TP53 mutant premalignant lesions. TP53 has long been recognized as a potential mediator of pregnancy-induced resistance to mammary carcinogenesis [47]. Later age at first pregnancy, and pregnancy by in vitro fertilization in premenopausal or menopausal aged women are associated with an increased possibility of PABC. PABC is considered rare, but its incidence shows an increasing trend in parallel to increasing age at childbirth [6,35].

Our analysis confirmed that PABC was associated with prognosis factors, similar to those reported among BC patients over 35 years of age. PABC is considered to have a worse outcome than BC in non-pregnant women, partly due to its intrinsic biological aggressivity (TNBC, HER2+). When diagnosed during pregnancy, BC exhibits aggressive behavior, more frequently occurring as triple-negative or HER2+ positive disease. Apart from the hormonal influences on mammary glands during pregnancy, we do not fully understand the interaction between pregnancy and breast cancer carcinogenesis. Studies have demonstrated that young age (under 35 years) is a negative prognostic factor for breast cancer. Compared to older patients, younger patients with breast cancer more commonly exhibit factors associated with local recurrence and worse survival—including non-luminal types, grade 3 tumors, lymphatic vessel invasion, necrosis, and lack of ER expression [50]. However, among total breast cancer deaths, 6.7% occur among women less than 45 years of age [51]. A multicentric analysis published in 2013 suggests that DFS and OS in BC patients under the age of 45 did not significantly differ between patients diagnosed and treated during pregnancy compared with non-pregnant patients when controlled for stage, and with a median follow-up of 61 months [22].

Additionally, PABC diagnosis can be delayed because it is a rare pathology that is not obvious for clinicians, and because diagnosis can be complicated due to physiological mammary modifications and limited investigation possibilities in pregnant women [52]. In our study, 58.33% (7/12) of cases were HER2+ and/or triple-negative. Of the 12 cases, 2 were diagnosed in the metastatic stage, and another 7 cases were locally advanced (T3–T4 and/or N1).

Since 2012, all PABC cases at our hospital have received a standard regimen of chemotherapy in the second and third trimester of pregnancy, resulting in incomplete remission. In cases of TNBC and HER2+ disease, pathologic complete response is correlated with DFS and OS [53]. However, although none of our patients achieved pCR, only one case exhibited recurrence (at five years). One possible explanation for the lack of pCR is that no dose-dense regimen was used in order to avoid the use of granulocyte colony-stimulating factor (G-CSF); although this is allowed by current guidelines [54]. For TNBC cases, NACT regimens did not include platinum-based therapy due to the limited data available at that time; however, guidelines now consider this approach acceptable [55].

Our present analysis could be biased by the different lengths of the follow-up periods (very short for recent patients), and by the differences in the therapeutic approach, which reflect updates of oncology guidelines. In one case (No. 9), we used an adjuvant treatment with capecitabine, which is reportedly associated with improved OS in non-pCR patients with TNBC after NACT [56]. In two cases (Nos. 11 and 12), we used a neoadjuvant treatment with trastuzumab and pertuzumab, instead of trastuzumab alone (the previous standard), to increase the pCR rate [57]; however, this regimen was not used for case No. 8 (not reimbursed at that time). Additionally, for one HER2+ patient (No. 12) with residual disease after NACT, we administered adjuvant treatment with trastuzumab emtansine (TDM1), which is associated with a decreased risk of recurrence and improved DFS [58]. This treatment was not available for the other HER2+ patients.

Guidelines recommend genetic testing for BRCA mutations in all breast cancer patients of <50 years. This test was only performed in 33% of patients in our series. The increased frequency of genetic mutation (VUS and BRCA1 mutations) in young BC patients may explain the different pathogenesis. Genetic testing of all patients can allow treatment personalization and improved outcomes [59].

Although cases of PABC are considered high-risk pregnancies with known potential complications (premature rupture of membranes and intrauterine growth restriction), our present series exhibited no obstetrical events or infant-related complications.

The two cases of maternal death can be explained by the very long delay of diagnosis and treatment, such that both cases showed advanced metastases at diagnosis and exhibited rapid general clinical deterioration. In the case lost to follow-up in February 2017, the patient was diagnosed during the postpartum period with invasive ductal carcinoma cT3N0 (ER−, PgR−, and HER2+) with liver, lung, and bone metastases. We obtained a partial response with first-line chemotherapy (docetaxel q3w and trastuzumab q3w). Grade 2 sensitive neuropathy occurred after 10 cycles of chemotherapy and taxane treatment; therefore, taxanes and anti-HER2 therapy were switched to capecitabine, trastuzumab, and pertuzumab. Targeting HER2, e.g., with transtuzumab and lapatinib, is a major step towards improvement of new cancer therapies, and is similar to targeting the estrogen receptor through hormone therapy.

Our present results are in accordance with prior studies, confirming that pregnancy itself does not seem to influence breast cancer outcome, with the therapeutic approach being similar between PABC and BC in non-pregnant patients, according to the disease type and stage [60]. All infants exhibited healthy development, and no cardiac side effects from chemotherapy were reported for either the mothers or infants.

Our study has several limitations. Our analysis includes only a small number of patients due to the rarity of PABC. INCIP register (including 37 centers from 16 countries) includes 462 cases of PABC reported between 2005 and 2020, representing an average of 28 cases per year, and 12 cases per center, over 16 years [35]. Additionally, a recent study including 11,846,300 deliveries between 1999 and 2012 reports an PABC incidence of 6.5 cases/100,000 pregnancies [61]. Our study only included patients who underwent chemotherapy at our hospital. Since the patients had their diagnoses established by the attending physician, we do not know the number of abortions performed upon request due to PABC. Moreover, this study was non-analytical; there were no control patients for cases, and the treatment protocols differed across the time period.

## 5. Conclusions

In conclusion, our present findings indicate that it is important for pregnant women to undergo clinical examination of the breasts, to promote timely detection of the rare pathology of PABC, especially in patients over 30 years of age. Pregnancy itself seems to be an independent factor affecting prognosis. Case management and prognosis in PABC are similar in non-pregnant BC patients, determined according to clinical stage, timely treatment, and patient age. Chemotherapy during pregnancy is allowed after a gestational age of 16 weeks. No intrauterine growth restriction or small-for-gestational age newborns were observed in our study. Due to the rarity of PABC, strengthening the means for a global partnership may improve PABC survival.

## Figures and Tables

**Figure 1 medicina-56-00522-f001:**
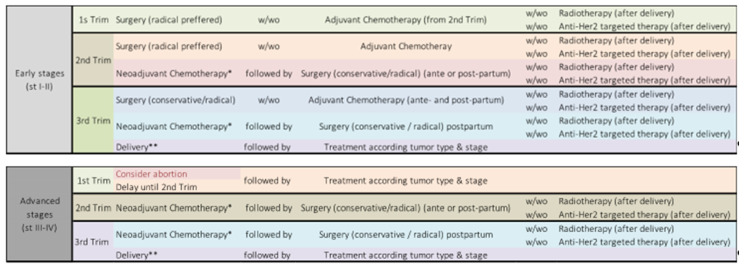
Treatment algorithm. * For triple negative breast cancer (TNBC), HER2+ breast cancer (BC), and some Luminal B cases. ** For a gestational age of ≥28 weeks. Trim = trimester.

**Table 1 medicina-56-00522-t001:** Characteristics of the studied population.

Characteristics		Number (N)	%
**Age at diagnosis**			
	16–25 yrs.	0	0
	26–30 yrs.	1	8.33
	31–35 yrs.	3	25.00
	36–40 yrs.	8	66.67
	>40 yrs.	0	0
**Gestational age at diagnosis**			
	1st Trim	1	8.33
	2nd Trim	5	41.67
	3rd Trim	4	33.33
	Post-partum	2	16.67
**Reproductive factors**			
	Menarche < 12 yrs	1	8.33
	Menarche > 12 yrs	11	91.67
	Spontaneous pregnancy	11	91.67
	FIV	1	8.33
	Primigravida	5	41.67
	Secundigravida	5	41.67
	Multigravida	2	16.67
	Time since last birth < 5 yrs.	7	58.33
	Time since last birth > 5 yrs.	5	41.67
**Maternal BMI**			
(kg/m^2^)	< 18.5	1	8.33
	18.5–24.9	6	50
	25.0–29.9	5	41.67
	>30	0	0
**Tobacco use**			
	Yes	0	0
	No	12	100
**Alcohol use**			
	Yes	0	0
	No	12	100
**Family history of cancer**			
	Yes	4	33.33
	No	8	66.67

**Table 2 medicina-56-00522-t002:** Cancer characteristics and management.

Case No.	Stage			Grade	Molecular Subtype	Gene Mut Status	Chemo during Preg	NACT	Response	Surgery	Adj Chemo	Adj RT	Adj HT	HER2 Treat	DFS (mo)	OS (mo)	Comments
	T	N	M														
1	2	1	0	3	TNBC	UNK	No	N/A	N/A	R	A, T	Yes	No	No	101	101	
2	4b	0	0	2	Luminal B	UNK	Yes	A,T	PR	R	No	Yes	Yes	No	66	98	A
3	2	1	0	x	HER2+	UNK	No	A	UNK	R	No	No	No	No	N/A	3	B
4	2	1	0	2	Luminal A	UNK	Yes	A,T	PR	R	No	No	Yes	No	77	85	
5	3	0	1	2	HER2+	UNK	No	N/A	N/A	N/A	T (M1 setting)	N/A	No	Yes (H adj)	N/A	9	C
6	4b	3	1	3	Luminal B	UNK	No	N/A	N/A	N/A	N/A	N/A	N/A	N/A	N/A	N/A	D
7	2	0	0	3	Luminal B	Neg	Yes	A,T	PR	R (+PCM)	No	No	Yes	No	28	36	
8	m2	0	0	2	HER2+	VUS (CDH1)	Yes	A,T	PR	R (+PCM)	No	No	Yes	Yes (H adj)	22	36	
9	2	1	0	3	TNBC	BRCA1 mut	No	A,T	N/A	R (+PCM)	Xel	No	No	No	24	32	
10	1	0	0	1	Luminal A	VUS (Rad50)	No	N/A	N/A	R	No	N/A	Yes	No	21	22	
11	3	0	0	2	HER2+	UNK	No	A,T	PR	R (+PCM)	No	No	Yes	Yes (H, P neoadj)	5	10	
12	2	1	0	3	HER2+	UNK	Yes	A,T	PR	R	No	Yes	No	Yes (H, P neoadj; TDM1 adj)	1	10	E

Chemo=chemotherapy; preg=pregnancy; mo=months; mut = mutational; NACT = neoadjuvant chemotherapy; adj = adjuvant; FIV = fertilization in vitro; DFS = disease-free survival; OS = overall survival; RT = radiotherapy; HT = hormonal therapy; TNBC = triple negative breast cancer; UNK = unknown; VUS = variant of unknown significance; N/A = not applicable; A = anthracyclines; T = taxanes; PR = partial response; R = radical; PMC = prophylactic contralateral mastectomy; H = trastuzumab (Herceptine); P = pertuzumab (Perjeta); TDM1 = trastuzumab emtansine (Kadcyla); BRA = brain; HEP = liver; OSS = bone; PUL = lung; BC = breast cancer; FU = follow-up. A: M1 BRA, HEP, OSS, PUL, controlat BC. B: Perioperative death. C: Lost to FU. D: Postpartum maternal death M1 SK BRA HEP PUL. E: Treatment ongoing.

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
