# Peer review of "Real-World Data Analysis of Pregnancy-Associated Breast Cancer at a Tertiary-Level Hospital in Romania"

_medicina, 2020, doi:10.3390/medicina56100522_

Round 1
Reviewer 1 Report
Good review and well presented protocols in use for the management of PABC.
In think readers will find this paper useful
One question: There is no mention of doing sentinel node biopsy why?
Comments about the safety of technetium or blue day should be included.
Some mention of monitoring the patient and the fetus during surgery by the anesthesiologist will be a good idea
Author Response
Please see the attachement

Reviewer 2 Report
Title: I suggest a title change. Real world seems like other authors are in irreal world
Introduction: Introduction is too longer. Very well explained and with excellent citations but not all the paragraphs are needed for article introduction. I suggest reducing the long and some issues may be discussed on the discussion section.
Material and methods: I miss to have some explanation about how obstetrical care was given during chemo treatment.
Results: Somme results are not well explained. Somme patients are not treated according to current guidelines. Number of patients is short and don’t show us new interesting data. Somme results must be better explained:
- Trastuzumab is started after delivery. Does any patient suffer a delay in her treatment due to that?
- Treatment is stopped at 35 weeks but according to the authors, induction was only because obstetrical issues. Does it mean that patients suffered a delay in scheduling her treatment?
- On the literature there is no reported cases of neonatal anemia or leucopenia because chemotherapy, so this can’t be a rationale for stopping treatment at 35 weeks.
- Median OS is short and not according to published data must be better explained in the results section and correctly explained on the discussion
- A Kapplan-Meier figure may help to understand survival of this cohort
- Mammography may be performed in pregnant women and nust be performed in all patients with cancer confirmation or suspicion.
- Delivery by C-section in all cases is not justified.
- There’s no information about newborns (weight, pH, NICU admissions)
Discussion: Authors had to explain why they find results so opposite to published data.
Author Response
Please see the attachment.

This manuscript is a resubmission of an earlier submission. The following is a list of the peer review reports and author responses from that submission.